# Lung Toxicity Occurring During Enfortumab Vedotin Treatment: From a Priming Case Report to a Retrospective Analysis

**DOI:** 10.3390/ph17111547

**Published:** 2024-11-18

**Authors:** Grégoire Desimpel, François Zammit, Sarah Lejeune, Guillaume Grisay, Emmanuel Seront

**Affiliations:** 1Institut Roi Albert II, Department of Medical Oncology, Cliniques Universitaires Saint-Luc, University of Louvain, 1200 Brussels, Belgium; gregoire.desimpel@student.uclouvain.be (G.D.); francois.zammit@student.uclouvain.be (F.Z.); 2Department of Medical Oncology, Hospital of Jolimont (HELORA Group), 7100 La louvière, Belgium; sarah.lejeune@helora.be (S.L.); guillaume.grisay@helora.be (G.G.)

**Keywords:** antibody-drug conjugate, enfortumab vedotin, pneumopathy, corticosteroids, reversible, enfortumab vedotin discontinuation

## Abstract

**Background/Objectives:** Enfortumab vedotin (EV) is an antibody-drug conjugate (ADC) that combines monomethyl auristatin E (MMAE), a potent cytotoxic agent, with a monoclonal antibody targeting Nectin-4. It has emerged as a promising therapy for metastatic urothelial carcinoma (mUC), either as monotherapy or in combination with pembrolizumab, improving significantly the overall survival of these patients. EV is associated with common adverse events, including skin reactions, glucose imbalance, and peripheral neuropathy, which are usually mild in severity and easily manageable. **Methods:** Following an initial case of pleuro-pneumopathy occurring in a patient treated with EV, we conducted a retrospective analysis of EV effects on pulmonary imaging. **Results:** In a cohort of 20 all-comers mUC patients, we identified three cases of potentially EV-related lung toxicity, resulting in a pleuro-pneumopathy rate of 15%. Two of these cases appeared highly symptomatic and required high steroid doses, with a rapid resolution of symptoms and normalization of radiological findings. In one patient, rechallenge of EV was associated with reoccurrence of pneumopathy. We described the clinical and radiological features of these cases, as well as their evolution after EV discontinuation and rechallenge. **Conclusions:** This case series underscores the importance of close pulmonary monitoring during EV treatment.

## 1. Introduction

Nectins are part of the calcium-independent immunoglobulin superfamily of cell adhesion molecules, comprising four members (Nectin-1 to Nectin-4). They play essential roles in cell–cell adhesion, cellular viability, immune modulation, and immune evasion. Notably, Nectin-4 is frequently overexpressed in several cancer types, including urothelial cancer (UC). This overexpression is associated with enhanced cell proliferation, survival, and angiogenesis, making Nectin-4 a promising antigen target for cancer therapies, such as antibody-drug conjugates (ADCs) [1,2]. Enfortumab Vedotin (EV) represents a significant breakthrough in the treatment of advanced UC, merging an anti-Nectin-4 monoclonal antibody with the potent chemotherapeutic monomethyl auristatin E (MMAE) [3,4]. The phase III EV301 trial, enrolling patients progressing after platinum-based chemotherapy and immune checkpoint inhibitors (ICIs), showed that EV treatment resulted in a significant improvement in progression-free survival (PFS), overall survival (OS), and objective response rate, compared to historical chemotherapy [5]. Based on these results, EV received European Medicines Agency (EMA) approval in 2022. More recently, the phase III EV302-KEYNOTE A039 trial evaluated the EV-pembrolizumab association in previously untreated mUC patients; EV-pembrolizumab was shown, compared to platinum-based chemotherapy, to double the median PFS (12.5 vs. 6.3 months, respectively; *p* < 0.001) and median OS (31.5 vs. 16.1 months, respectively; *p* < 0.001) appearing now as a new standard practice in the first-line metastatic setting of UC [6]. This may be explained by the fact that, in preclinical studies, EV was shown to enhance the response to ICI by upregulating major histocompatibility complex (MHC) genes, by stimulating the maturation of macrophages and dendritic cells, and by increasing cytokines involved in T-cell activation [7]. Furthermore, an ongoing clinical trial is evaluating EV-pembrolizumab combination in the perioperative setting in localized UC (NCT03924895).

The majority of patients treated with EV may experience adverse events (AEs), such as alopecia, neuropathy, skin toxicities, and asthenia [5,6]. Although most of these AEs are mild to moderate in severity, some patients experience serious AEs, including rash, peripheral neuropathy, and hyperglycemia. Conversely, lung toxicity occurring during EV treatment is rarely reported; in the EV301 trial, one patient died due to pneumonia that was considered as potentially EV-related AE but without ruling out other causative agents [5]. A retrospective analysis including 64 Korean patients included in EV201 and EV301 trials reported a 25%-rate of EV-related pneumonitis [8]. With the increasing use of EV in mUC, it is crucial to recognize the potential for lung toxicity associated with EV, as well as the optimal management. In this analysis, we retrospectively reviewed the occurrence of pleuro-pneumopathy in patients treated with EV, detailing the clinical and radiological features, as well as the pulmonary evolution of these patients following EV interruption/rechallenge.

## 2. Results

### 2.1. Case 1 (Priming Patient) 

A 71-year-old male long-term smoker was diagnosed with localized UC of the right kidney (cT3N0M0) in January 2022. After four cycles of neoadjuvant cisplatin-based chemotherapy, nephro-ureterectomy revealed persistent high-grade UC (ypT2 pN0). In September 2022, pembrolizumab 200mg q3w was initiated for disease recurrence with retroperitoneal lymph nodes and liver metastases. Due to liver progression, pembrolizumab was interrupted after four cycles and paclitaxel (90 mg/m^2^, Day 1, 8, 15 in a 21-day cycle) was started but stopped after two cycles due to disease progression. In January 2023, EV was initiated at a standard dose (1.25 mg/kg, day 1, 8, 15 in a 28-day cycle). At this time, the patient did not report any pulmonary symptoms and a thoracic CT did not show any lung anomaly. However, after two cycles and 10 days after the last EV administration, the patient presented in emergency (day 0) for rapidly progressive dyspnea over five days, occurring at rest and requiring oxygen (2 mL/min) for a blood saturation not exceeding 89% (Grade 3 dyspnea, common terminology criteria for adverse events (CTCAE) v5.0). The patient did not report any temperature or cough. A thoraco-abdominal CT scan revealed diffuse and bilateral sub-pleural reticulations on the CT scan, while all metastatic lesions showed important regression in size; this bilateral subpleural thickening demonstrated an intense and homogenous fluorodeoxyglucose (FDG) fixation on PET-CT (Figure 1A,B). Laboratory analyses showed only a mild leucocytosis and a moderate inflammatory syndrome. Sputum culture, blood culture, and serologies (Cytomegalovirus, Ebstein Barr Virus, Mycoplasma Pneumoniae, Chlamydia pneumoniae, Legionella) were negative; COVID-19 was not detected on polymerase chain reaction (PCR). Due to the absence of bacterial infection proof, no antibiotic course was initiated, but intravenous methylprednisolone 1 mg/kg/day was started on day 3 for three days, which resulted in rapid dysnea improvement and oxygen cessation. Methylprednisolone was rapidly tapered over three weeks and EV was not continued. Six weeks later, a PET-CT showed a disappearance of bilateral sub-pleural FDG fixation but apparition of new metastatic lesions (Figure 1C,D). Best supportive cares were initiated and the patient died six weeks later due to progressive disease.

### 2.2. Retrospective Analysis

This “priming case” led us to review the thoracic imaging of all patients treated with EV in our institution. Twenty patients treated with EV were included in our retrospective analysis. All these patients were included because they were treated with EV and have available CT imaging before and after EV treatment. Lung CT scans were analyzed in all these patients. Including our priming patient, three patients from these 20 patients presented lung anomalies appearing during EV treatment, resulting in a 15% rate of pneumopathy that might be related to EV. The baseline characteristics of these three patients with lung anomaly occurring on EV are described in Table 1. The radiological patterns on EV treatment and after EV interruption are presented in Figure 2 and Figure 3.

Case 2. A 73-year male patient, former smoker, and with a severe broncho-emphysematous disease, presented with a retroperitoneal lymph node recurrence of a UC previously treated with cystectomy. This patient had chronic dyspnea at mild effort but did not require oxygen. Four cycles of carboplatin-based chemotherapy resulted in stable disease, followed by maintenance avelumab (800 mg q2w). However, four weeks after the fifth administration of avelumab, EV was initiated due to lymph node progression. At this time, the patient did not note any worsening of pulmonary function and the CT scan showed no pneumopathy, only stable emphysematous changes (Figure 2A). After one cycle of EV, a thoraco-abdominal CT showed a regression of lymph nodes but revealed a reticular abnormality in the right upper lobe, which remained asymptomatic. EV was continued at standard dose (Figure 2B). After completing two EV cycles, the patient developed a symmetrical drug-related intertriginous and flexural exanthem (SDRIFE) and worsening dyspnea over 5 days, which appeared at rest and required oxygen (grade 3 dyspnea CTCAE v5.0). There was no history of cough or temperature. The thoracic CT scan revealed an increase in the right lung condensation (Figure 2C). The absence of bacterial suspicion and the concurrent EV-related AE led to initiation of methylprednisolone (1 mg/kg intravenously daily) for three days, resulting in rapid improvement in respiratory symptoms and skin lesions. Methylprednisolone doses were tapered over three weeks and EV was discontinued. Two months after EV interruption, the CT scan showed the disappearance of lung condensation, as well as an oncological stability (Figure 2D). However, two months later, the progression of lymph nodes on the CT scan led to EV reintroduction at a lower dose (1.0 mg/kg at day 1, 8, 15 on a 28-day cycle). Interestingly, skin toxicity reappeared rapidly during the first cycle but was successfully managed with local treatment. A PET-CT performed two months after EV reintroduction showed an oncological partial response but increased lung condensation, which demonstrated a FDG fixation on the PET CT (Figure 2E). EV was subsequently interrupted, leading to a decrease in lung condensation two month later (Figure 2F).

Case 3. A 68-year-old male patient was diagnosed with UC of the kidney associated with lombo-aortic and mediastinal lymph nodes. He had a long history of smoking but recently arrested. Four cycles of cisplatin-gemcitabine resulted in partial response and maintenance avelumab (800 mg q2w) was given. Four weeks after the fifth dose of avelumab, EV was started at a standard dose due to locoregional and mediastinal progression. He was, at this time, asymptomatic and the CT scan did not reveal any lung anomaly (Figure 3A). After completing three EV cycles, the CT scan demonstrated a partial response based on RECIST 1.1 criteria. While our patient was asymptomatic, thoracic imaging revealed sub-pleural arciform consolidations and ground-glass opacities (Figure 3B). EV was interrupted and close follow-up was started. Two months later, the CT scan showed decreased sub-pleural ground-glass opacities and stability of metastatic lesions, which was the reason why we did not reintroduce EV (Figure 3C). Although still present, these pulmonary anomalies decreased on the CT scan performed four months after EV arrest (Figure 3D). However, due to disease progression two months later, EV was restarted at a lower dose (1.0 mg/kg on days 1, 8, 15 of a 28-day cycle). Three months later, the patient remains on this regimen, with a partial response and no pneumopathy observed on the CT scan.

## 3. Discussion

Pleuro-pneumopathy has not traditionally been highlighted as a primary concern associated with EV monotherapy. However, with the increasing use of EV, it is crucial to document all potential EV-related pulmonary toxicities to refine our follow-up protocols and management strategies. In our cohort of 20 patients, three (15%) developed pneumopathy during EV treatment, suggesting a potential link between EV and pulmonary toxicity. These findings are consistent with those reported by Yoon et al. in their retrospective analysis of 64 Korean patients from the EV-201 and EV-301 trials [5].

Although our report covers only three patients, we provide a detailed account of the clinical and radiological characteristics of lung anomalies occurring during EV treatment and their evolution after EV interruption. The clinical manifestations of these pneumopathies were rapidly progressing and severe in two patients, requiring oxygenotherapy, without any announcing symptoms such as temperature or cough. The involvement of EV in these pneumopathies appears significant for several reasons. (1) Temporal onset and resolution: pneumopathy emerged rapidly after the initiation of EV, with an onset between 6 and 12 weeks, and resolved quickly after EV was discontinued. In one patient, reintroduction of EV led to a recurrence of pneumopathy. (2) Concurrent toxicities: in one patient, the onset of pneumopathy coincided with EV-related skin toxicity, further implicating EV in these lung anomalies. (3) Lung anomalies emerged while metastases appeared controlled, ruling out lung carcinomatosis. Furthermore, the resolution of pneumopathy with corticosteroid treatment alone, without antibiotic, and following EV interruption, is consistent with a drug-induced lung disease.

These EV-related pleuro-pneumopathies displayed atypical radiological features, such as reticulation, ground-glass opacities, and subpleural infiltrations, with a pronounced inflammatory component. This was evidenced not only by PET-CT imaging but also by the positive response to corticosteroid treatment. Interestingly, the extent of radiological anomalies was relatively moderate compared to the severity of symptoms observed in two patients. Although functional respiratory tests were not performed, they should have been included in the assessment. There is growing evidence that EV can enhance the efficacy of immunotherapy drugs by stimulating immunogenic cell death [9]. However, it remains unclear whether the described pneumopathies are solely related to EV or are a delayed effect of prior ICI therapy, potentially facilitated by EV, as all our patients had previously received ICIs with a short interval between ICI cessation and EV initiation. This distinction is particularly important as EV-pembrolizumab combination therapy is expected to become the standard of care in the first-line metastatic setting. Differentiating between pulmonary toxicities related to EV and those induced by ICI will be critical. Additionally, it is unknown whether EV might increase the likelihood of complications from COVID-19 or other viral infections. In our patients, no recent (≤6 months) COVID-19 infection or vaccination was reported [6].

The optimal management of EV-related pneumopathy remains uncertain. We did not administer antibiotics due to the absence of clinical signs or a radiological aspect supporting bacterial infection. In the two symptomatic patients, intravenous corticosteroids (1 mg/kg) led to rapid symptom improvement and reduced oxygen dependency. In the asymptomatic patient, discontinuation of EV alone resulted in a decrease in radiological anomalies.

Drawing from our experience and existing recommendations for managing drug-induced pneumonitis [10,11], such as those related to ICI or the ADC trastuzumab-deruxtecan, we propose a therapeutic strategy for addressing pneumopathy associated with EV (see Figure 4). The decision to reintroduce EV at either the standard or a reduced dose remains challenging; in our cases, a dose reduction was preferred, although this did not prevent the recurrence of lung anomalies in one patient, emphasizing the need for close follow-up.

Limitations of this report are the very low number of patients that are included in this study, and the retrospective analysis of patient evolutions.

## 4. Materials and Methods

In Belgium, EV monotherapy has been offered to patients with mUC after failure of chemotherapy and ICI, since December 2022 (initially offered under compassionate use, it became reimbursable in March 2023). One of our patients treated with EV developed severe pneumopathy early in the course of treatment. This case is described in this manuscript as the “priming” patient. Following this case, we retrospectively analyzed a cohort of all-comers patients treated with EV monotherapy for mUC from January 2023 to March 2024 in two Belgian institutions (Cliniques Universitaires Saint-Luc and Groupe Jolimont) regardless of the presence/absence of pulmonary symptoms; lung computed tomography (CT) of these patients was analyzed in order to identify the emergence of lung anomalies potentially induced by EV. To be included in this analysis, patients needed to have a lung CT performed within the four weeks before EV initiation (baseline imaging) and during EV treatment on a 3- or 6-month base. Patients with pre-existing pneumopathy (before EV treatment) were excluded. All the CT scan imaging performed after EV initiation was retrospectively reviewed and compared to baseline imaging. Pneumopathy was defined as the occurrence of new and diffuse lung parenchymal abnormalities, including ground-glass opacities, consolidations, and reticulations. We considered pneumopathy that might be related to EV if there was: (1) a temporal association with EV initiation, with absence of lung anomalies before starting EV; (2) an exclusion of other etiologies; and (3) an improvement after discontinuation of EV. The percentage rate of pneumopathy induced by EV was calculated as the number of patients presenting pneumopathy during EV treatment, divided by the number of patients treated with EV.

This retrospective analysis was approved by ethical committee from the two centers.

## 5. Conclusions

This case series underscores the necessity to closely monitor patients on EV, given the risk of associated pulmonary toxicity. However, more extensive investigations are warranted, focusing on larger patient cohorts and extended follow-up durations. Future prospective studies should investigate the real incidence of this toxicity by systemic appropriate imaging tools, as well as the optimal management with steroids. Potential biomarker for toxicities should also be evaluated; for instance, it would be interesting to assess the expression levels of Nectin-4 in lung parenchyma to better understand the mechanisms behind antibody-drug conjugate toxicity.

## Figures and Tables

**Figure 1 pharmaceuticals-17-01547-f001:**
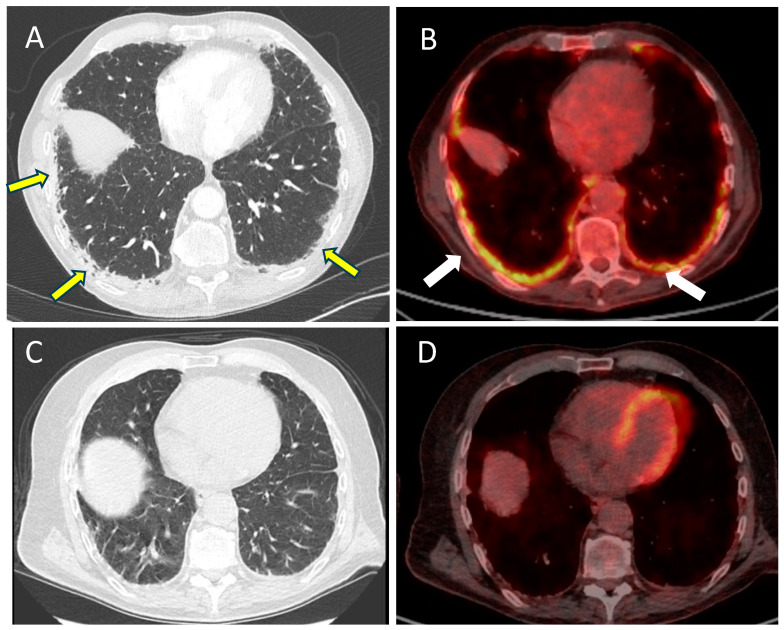
(**A**,**B**): Imaging post-EV initiation. Bilateral sub-pleural reticulate infiltration (yellow arrow) on CT scan (**A**), demonstrating a fluorodeoxyglucose fixation (white arrow) on PET-CT (**B**). (**C**,**D**): Imaging post-corticosteroids and EV interruption. Decrease in sub-pleural thickening on CT scan (**C**) and metabolic fixation (**D**).

**Figure 2 pharmaceuticals-17-01547-f002:**
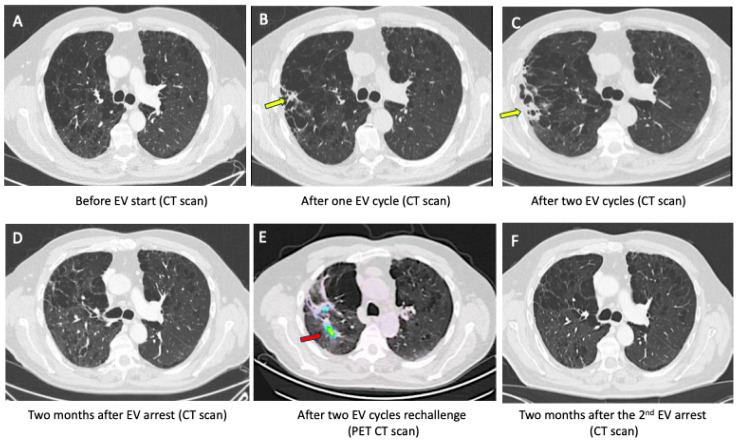
Patient 2: Baseline lung CT before EV initiation (**A**). Sub-pleural condensation (yellow arrow) appearing after one (**B**) and two cycles of EV (**C**). Decrease in sub-pleural condensation on CT scan two months after EV interruption (**D**). Re-emergence of lung condensation which demonstrated FDG fixation on PET CT (red arrow) performed two months after EV reintroduction (**E**). Interruption of EV resulted again in lung condensation disappearance (**F**).

**Figure 3 pharmaceuticals-17-01547-f003:**
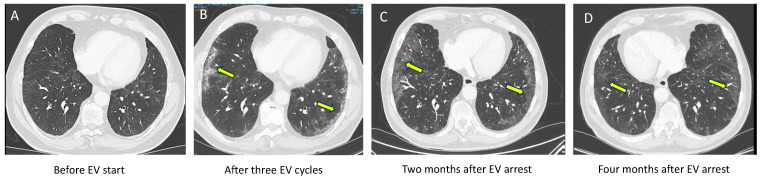
Patient 3: Baseline lung CT before EV initiation (**A**). Sub-pleural and parenchymal anomalies (yellow arrows) after three cycle of EV (**B**) and their evolution (yellow arrows) after EV interruption (**C**,**D**).

**Figure 4 pharmaceuticals-17-01547-f004:**
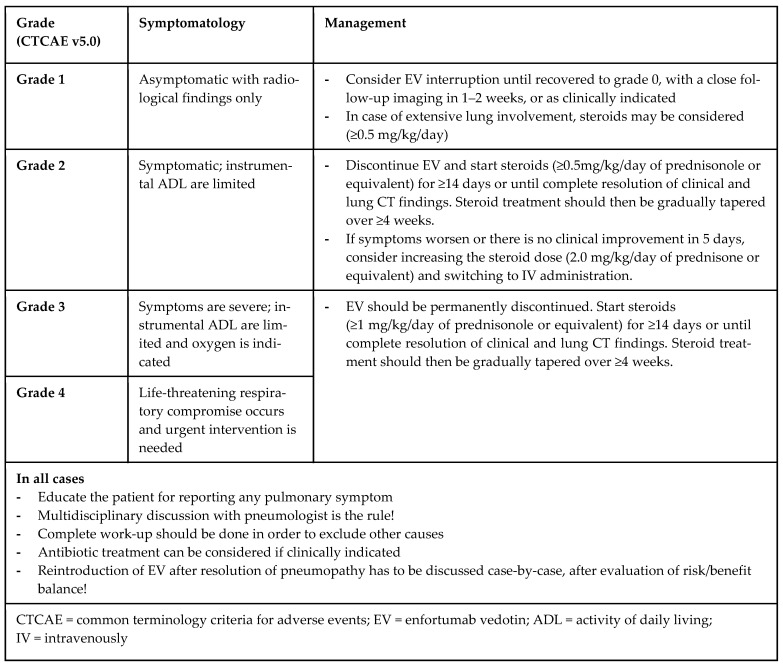
Proposed management for pneumopathy occurring during enfortumab vedotin treatment.

**Table 1 pharmaceuticals-17-01547-t001:** Clinical characteristics of the three patients who experienced lung toxicity induced by enfortumab vedotin.

	Patient 1	Patient 2	Patient 3
Sex	Male	Male	Male
Age at EV initiation (years)	71	73	68
Risk factors	Current smoker	Former smoker	Former smoker
	40-packs year	20-packs year	25-packs year
Previous chemotherapy			
- setting	Neoadjuvant	Metastatic	Metastatic
- type	Cis-Gem	Carbo-Gem	Cis-Gem
- number of cycles	4	4	4
Previous ICI			
- setting	Metastatic	Maintenance	Maintenance
- agent	Pembrolizumab	Avelumab	Avelumab
- duration (weeks)	12	10	10
- pulmonary toxicity related to ICI	None	None	None
EV treatment			
- interval between EV and ICI (weeks)	8	4	4
- Interval EV initiation and lung tox (weeks)	8	6	12
- Dose of EV	1.25 mg/kg	1.25 mg/kg	1.25 mg/kg
- Associated toxicity	None	Skin toxicity	None
Lung metastases at EV initiation	No	Yes	No
Pulmonary symptoms	- No temperature - No cough - Reaching Grade 3 dyspnea over 5 days - Oxygen needed	- No temperature - No cough - Reaching Grade 3 dyspnea over 5 days - Oxygen needed	- No temperature - No cough - No dyspnea
Laboratory anomalies			
- Leucocytosis (/mm^3^)	12,000	10,000	8000
- Neutrophiles %	65	71	60
- Lymphocytes %	20	14	33
- CRP (mg/dL)	70	90	25
- Recent COVID infection <6 months	None	None	None
- Recent COVID vaccination <6 months	None	None	None
- PCR COVID	Not detected	Not detected	Not detected
Treatment			
- Antibiotic	No	No	No
- Methylprednisone	1 mg/kg IV for 3 days, tapered over 3 weeks	1 mg/kg IV for 3 days, tapered over 3 weeks	No
EV reinitiation	No	Yes, at lower dose	Yes, at lower dose
Lung tox recurrence	No	Yes	No

EV = enfortumab vedotin; ICI = immune checkpoint inhibitors; Cis-Gem = cisplatin-gemcitabine; carbo = carboplatin; grade dyspnea following common terminology criteria for adverse events (CTCAE v5.0); IV = intravenously.

## Data Availability

Data are available on demand.

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
