# Peer review of "Lung Toxicity Occurring During Enfortumab Vedotin Treatment: From a Priming Case Report to a Retrospective Analysis"

_pharmaceuticals, 2024, doi:10.3390/ph17111547_

Round 1
Reviewer 1 Report
Comments and Suggestions for Authors
Dear Authors,
The manuscript presents an interesting study that highlights possible adverse reactions to a current therapeutic strategy in the pathology of urothelial carcinoma.
In support of this manuscript for publication in the journal Pharmaceuticals, some suggestions are needed on which the authors should provide some additional clarification.
Lines 30-31 I suggest the authors to add additional explanations related to both the pathology (in a few lines the pathology of advanced urothelial carcinoma, spread, etc. considering that it is the tenth most common type of cancer worldwide) and the medication addressed, recent therapeutic strategies: association of anti-nectin-4 monoclonal antibody with monomethyl auristatin E (MMAE):
- The etiology of the nectin-4 protein: the structure and importance of the four types of nectins. Nectin-4 being a tumor-associated antigen found on the surface of most urothelial carcinoma cells. In the antibody-drug conjugate enfortumab vedotin, the human anti-nectin-4 antibody is linked to the cytotoxic microtubule-disrupting agent monomethyl auristatin E.
- the advantages of this association.
The addition of bibliographic titles related to urothelial carcinoma and nectin-4 targeting strategies – see article published in 2021 in Nature Reviews Urology volume 18.
Line 61 - the authors mention that a cohort study was initiated between January 2023 and March 2024 in patients treated as monotherapy with Enfortumab Vedotin. For clarity, I suggest mentioning here the number of patients monitored during this period.
This analysis of the adverse events of lung damage is important, a percentage of 15% of the patients included in the study is relevant and important considering the severity of the pathology associated with EV administration. Thus, knowing this impairment during treatment with EV, preliminary measures can be taken to avoid complications, which highlights the importance of this analysis. Also highlighting the limitations of treatment with EV is remarkable, as well as the proposal of the study team on the therapeutic conduct and adequate counseling of the patient treated with this medicine.
In the experimental part of the work, it is necessary to use the appropriate names for chemicals and reagents (composition, concentration and manufacturer), as well as for apparatus (type, manufacturer).
The manuscript is well written, the analysis is compelling, and the limitations are well identified. Therefore, I encourage the authors to conduct further studies to evaluate the safety profile of this association by extending the studies to a larger number of patients.
Author Response
The manuscript presents an interesting study that highlights possible adverse reactions to a current therapeutic strategy in the pathology of urothelial carcinoma.
In support of this manuscript for publication in the journal Pharmaceuticals, some suggestions are needed on which the authors should provide some additional clarification.
Lines 30-31 I suggest the authors to add additional explanations related to both the pathology (in a few lines the pathology of advanced urothelial carcinoma, spread, etc. considering that it is the tenth most common type of cancer worldwide) and the medication addressed, recent therapeutic strategies: association of anti-nectin-4 monoclonal antibody with monomethyl auristatin E (MMAE):
- The etiology of the nectin-4 protein: the structure and importance of the four types of nectins. Nectin-4 being a tumor-associated antigen found on the surface of most urothelial carcinoma cells. In the antibody-drug conjugate enfortumab vedotin, the human anti-nectin-4 antibody is linked to the cytotoxic microtubule-disrupting agent monomethyl auristatin E.
 thank you for this comment. We added a paragraph in the introduction concerning Nectin-4
“Nectins are part of the calcium-independent immunoglobulin superfamily of cell adhesion molecules, comprising four members (Nectin-1 to Nectin-4). They play essential roles in cell-cell adhesion, cellular viability, immune modulation, and immune evasion. Notably, Nectin-4 is frequently overexpressed in several cancer types, including urothelial cancer. This overexpression is associated with enhanced cell proliferation, survival, and angiogenesis, making Nectin-4 a promising antigen target for cancer therapies, such as antibody-drug conjugates (ADCs) (1)”
- the advantages of this association.
 we added also a paragraph to answer this query
“This may be explained by the fact that, in preclinical studies, EV may enhance the response to ICI by upregulating Major Histocompatibility Complex genes, by stimulating the maturation of macrophages and dendritic cells, and by increasing cytokines involved in T-cell activation (6).”
The addition of bibliographic titles related to urothelial carcinoma and nectin-4 targeting strategies – see article published in 2021 in Nature Reviews Urology volume 18.
 Added
Line 61 - the authors mention that a cohort study was initiated between January 2023 and March 2024 in patients treated as monotherapy with Enfortumab Vedotin. For clarity, I suggest mentioning here the number of patients monitored during this period.
 Only twenty patients were monitored
This analysis of the adverse events of lung damage is important, a percentage of 15% of the patients included in the study is relevant and important considering the severity of the pathology associated with EV administration. Thus, knowing this impairment during treatment with EV, preliminary measures can be taken to avoid complications, which highlights the importance of this analysis. Also highlighting the limitations of treatment with EV is remarkable, as well as the proposal of the study team on the therapeutic conduct and adequate counseling of the patient treated with this medicine.
In the experimental part of the work, it is necessary to use the appropriate names for chemicals and reagents (composition, concentration and manufacturer), as well as for apparatus (type, manufacturer).
The manuscript is well written, the analysis is compelling, and the limitations are well identified. Therefore, I encourage the authors to conduct further studies to evaluate the safety profile of this association by extending the studies to a larger number of patients.
 Thank you
Reviewer 2 Report
Comments and Suggestions for Authors
The authors present a study detailing a priming case as well as three cases of mUC patients in a retrospective analysis suffering from lung toxicity after treatment with the ADC Enfortumab vedotin (EV), providing anecdotical evidence suggesting that lung toxicity is a frequent adverse effect of EV treatment of lung cancer patients.
The analysis is well conducted with a rather comprehensice clinical description of the patient cases resulting in sound conclusions concerning the possible contribution of previous ICI and EV treatment to the observed lung toxicity.
The treatment of the patients is representative of the standard cancer therapy, relying on ICI and immunotherapy often requiring the use of additional drugs to mitigate adverse effects of the cancer therapy, akin to or representing polypharmacy which is cautiously discussed by the authors. In this respect, the avoidance of antibiotic treatment and the absence of COVID-19 mRNA vaccination in the case reports as important AE sources are beneficial for the conclusions.
Some amendments are recommended:
Since the lung toxicity AE reflects the rather unsatisfactory success rates of the standard oncology therapies, often accompanied by severe toxicity and subsequent damage, alternative low-risk measures as combined therapies should be mentioned: For example, plant nutrients as part of the diet, such as ginger derivatives are of substantial interest providing direct anti-tumour effects as well as being chemopreventive, thus mitigating AEs from ICI, etc. (for review: Zadorozhna and Mangieri. Mechanisms of Chemopreventive and Therapeutic Proprieties of Ginger Extracts in Cancer. Int. J. Mol. Sci. 2021, 22, 6599. https://doi.org/10.3390/ijms22126599). Similarly, high doses of properly administered ascorbic acid are of interest (E.g. for review: Isola et al. Vitamin C Supplementation in the
Treatment of Autoimmune and Onco-Hematological Diseases: From Prophylaxis to Adjuvant Therapy. Int. J. Mol. Sci. 2024, 25, 7284. https://doi.org/10.3390/ijms25137284). Furthermore, simple but effective measures addressing metabolic peculiarities of tumor cells, e.g. fasting and strict avoidance of sugar uptake (i.e. cutting off tumor cells from glucose as an essential nutrient) or increasing the pH level in the TME (as tumors can only grow in an acidic environment) are of interest.
These measures are also considerable as alternative/supplement to minimise AEs from the standard steroid treatment to cope with lung toxicity.
Author Response
 Thank you for your comment; they are really interesting and may be a subject for a separate article. We prefer to not add such paragraph in the manuscript as this is only a description of an adverse event. The role of alternative low-risk measures as suggested is really unknown, as well as the interactions they can have with new compounds (Enfortumab Vedotin)
Reviewer 3 Report
Comments and Suggestions for Authors
Please check attached comments document.

Please check comments to authors.
Author Response
1) in the abstract is well presented the Research aims but it is suggested to add some insights of results in order to wake up the interest of readers to go through the article.
 Thank you for this comment. We added a sentence in the abstract
“Following an initial case of pleuro-pneumopathy occurring in a patient treated with EV, we conducted a retrospective analysis of EV effects on pulmonary imaging. In a cohort of 20 all-comers mUC patients, we identified three cases of potentially EV-related lung toxicity, resulting in a pneumopathy rate of 15%. Two of these cases appeared highly symptomatic and required high steroid doses.”
2) in the introduction please consider to rephrase the redaction in the end to add aims of Research at the place to incorporate soon results. It is a suggestion.
 You right; we adapted the end of introduction
“In this analysis, we retrospectively review the occurrence of pleuro-pneumopathy in patients treated with EV, detailing the clinical and radiological features, as well as the pulmonary evolution of these patients following EV interruption / rechallenge.”
3) please check the required format; and add a section of "materials and methods".
 adapted
4) in the material and methods add ethical issues considered to develop the Research.
 There was no ethical issue as it is a descriptive retrospective analysis and the approval of ethical committee is present in the materials and methods
5) about the results and way to present them and analyze them too. It is suggested to revise the way of analyzing the case as whole of source of data from all patients as stated from the begining to detect the case to highlight. Then, develop the retrospective analysis to understand the causes depending of the aims.
 we tried to write the manuscript I this way, beginning from the “priming” case and then describe the retrospective analysis. We tried to render the manuscript more clear. For example, we added a transition sentence before the “retrospective analysis” paragraph, such “This “priming case” led us to review the thoracic imaging of all patients treated with EV in our institution.”
Round 2
Reviewer 3 Report
Comments and Suggestions for Authors
Thank you very much for your responses and modifications related. Now. from the Reviewer side, it is accepted to continue with the Editorial process to publish it.
Comments on the Quality of English LanguageNo apply